# Neodymium Selenide Nanoparticles: Greener Synthesis and Structural Characterization

**DOI:** 10.3390/biomimetics7040150

**Published:** 2022-10-03

**Authors:** Abu A. Ansary, Asad Syed, Abdallah M. Elgorban, Ali H. Bahkali, Rajender S. Varma, Mohd Sajid Khan

**Affiliations:** 1Unique Sixth Form, 1-19 Wakefield Street, London N18 2BZ, UK; nmansary@gmail.com; 2Botany and Microbiology Department, College of Science, King Saud University, P.O. Box 2455, Riyadh 11451, Saudi Arabia; aelgorban@ksu.edu.sa (A.M.E.); abahkali@ksu.edu.sa (A.H.B.); 3Regional Centre of Advanced Technologies and Materials, Czech Advanced Technology and Research Institute, Palacký University in Olomouc, Šlechtitelů 27, 783 71 Olomouc, Czech Republic; varma.rajender@epa.gov; 4Nanomedicine & Nanobiotechnology Lab, Department of Biosciences, Integral University, Lucknow 226026, India

**Keywords:** biomimetic synthesis, Nd_2_Se_3_, synthetic peptide, structural characterization

## Abstract

This investigation presents the greener biomimetic fabrication of neodymium selenide nanoparticles (Nd_2_Se_3_ NPs) deploying nitrate-dependent reductase as a reducing (or redox) agent, extracted from the fungus, *Fusarium oxysporum*. The Nd_2_Se_3_ NPs, with an average size of 18 ± 1 nm, were fabricated with the assistance of a synthetic peptide comprising an amino acid sequence (Glu-Cys)n-Gly, which functioned as a capping molecule. Further, the NPs were characterized using multiple techniques such as UV-Vis spectroscopy, fluorescence, dynamic light scattering (DLS), and XRD. The hydrodynamic radii of biogenic polydispersed Nd_2_Se_3_ NPs were found to be 57 nm with PDI value of 0.440 under DLS. The as-made Nd_2_Se_3_NPs were water-dispersible owing to the existence of hydrophilic moieties (-NH_2_, -COOH, -OH) in the capping peptide. Additionally, these functionalities render the emulsion highly stable (zeta potential −9.47 mV) with no visible sign of agglomeration which bodes well for their excellent future prospects in labeling and bioimaging endeavors.

## 1. Introduction

Chalcogenides (compounds and alloys of sulfur, selenium, and tellurium) are known for their splendid physical properties as they have phenomenal magnetic, electronic, catalytic, sensing, thermal, optical, and superconductivity properties. The conversions of chalcogenides into nanocrystalline metal chalcogenides have revolutionized the importance of these materials [1]. In particular, semiconductor nanocrystals have attracted global consideration owing to their distinctive optical features, such as great resistance to photo bleaching, an enormous absorption cross-section, extended fluorescence lifetimes, good quantum yield, and a luminescence emission with a large Stokes shift [2]. Furthermore, their perceived properties make them a subject of study for biologically relevant applications, which extend to medicine and bioimaging [3]. Highly functional nanostructured materials are grown using a biological system (including silica via diatoms, and magnetic nanomaterials using magnetotactic bacteria) [4], among numerous ongoing efforts to assemble inorganic nanoparticles (NPs) by biological means thus mimicking such phenomena at a laboratory scale. Various organisms (such as bacteria, fungi, and actinomycetes) have been studied for the generation of NPs over a range of chemical compositions that include metals, semiconductors, and oxides [5,6,7,8,9].

Generally, semiconductor materials are defined by their composition-dependent band gap energy (Eg), which is the minimal energy needed to stimulate an electron from the valence band (VB) into the vacant conduction energy (CE) band [10]. Band gaps play substantial role in determining the purity of nanomaterials as well as in designing the fabrication of devices (sensors and photocatalysts) [11]. Other vital factors that may also affect the development of nanostructures in semiconductor NPs, include the comparative surface energies of the constituent metals, their corresponding rates of surface diffusion, and the conditions prevailing during or after deposition, whereas the large surface area enhances the surface properties. Additionally, these factors make whole-structure characterization important for these systems; moreover, they provide a more effective way to tune the structural morphology, and its corresponding properties. The relative differences between the energetic and fundamental structural parameters of a constituent’s metal and nonmetal vary considerably between elemental systems. For the last two decades, the applications of nano-chalcogenides have garnered worldwide attention and their nanocrystalline forms have received immense significance in various applications [12]. Therefore, different routes for the synthesis of nanocrystalline metal chalcogenides have evolved for the preparation of nano-chalcogenides of desired physical properties [13,14,15]. Metal chalcogenides can easily be tuned, and their optoelectronic, physicochemical, magnetic, and biological properties can conversely manipulate for energy to biomedical applications. Neodymium oxides nanomaterials are being extensively used in optical, antireflection coatings, gate insulators, protective coatings photonic, catalytic, and many special applications [9]. Inner transition metals nanomaterials have attractive luminescent properties. The promising utilities in time resolved luminescence bioassays of lanthanide compounds propose exceptional sharp fluorescence with highly distinguishable long lasting emission bands and large Stoke shifts [16]. Their luminescence is created by transition within 4f shell. To the best of our knowledge, this is the first ever fungal-derived protein-mediated biosynthesis of Nd_2_Se_3_ NPs which may help open new dimensions of fluorescence applications in nanomedicine.

Chemical and physical methods produce highly toxic, unsafe, environmental unfriendly, cumbersome, unstable colloidal NPs that also require tedious derivatization protocols. In this regard, it is highly necessary to develop high-quality semiconductor nanomaterials via cost-effective, convenient, and less toxic methods. Therefore, nanotechnologists are looking toward greener machineries to produce nontoxic, stable, and long-lasting nanomaterials that function under ambient conditions. To that end, various methodologies have been designed to produce biogenic nanomaterials. In addition to this, the green synthesis of Au, Ag, and CdS NPs have also been achieved using enzymes and peptides purified from fungus [17,18,19]. These synthesized NPs (green synthesis by enzyme) are capped with small metal binding peptides with flanking free carboxyl or amino groups [19]. These readily available functionalities obviate the need for biofunctionalization, which is required for conjugating the NPs to the biomolecules [19]. Particularly, a wide range of templates has been engaged for enzyme immobilization, namely silica nanotubes [20], phospholipid bilayers [21], and self-assembled monolayers [22]. Furthermore, numerous studies have been reported on semiconductor–enzyme conjugates that embrace the development and enzyme-mimicking behavior of NPs complexed with horseradish peroxidase [23], xanthine oxidase [24], and carbonic anhydrase [25].

Herein, our study presents a detailed investigation into the use of enzymes (e.g., nitrate reductase) cleansed from fungus (e.g., *Fusarium oxysporum*) for the fabrication of technically vital, highly stable semiconductor Nd_2_Se_3_ NPs capped with a synthetic peptide. We have observed that the aqueous Nd^2+^ and Se^4+^ ions were simultaneously oxidized and reduced (redox reaction), respectively with the aid of enzyme nitrate reductase, leading to the creation of stable Nd_2_Se_3_ NPs in solution; ensued NPs were in the range of 16–27 nm in size, with an average of 18 ± 1 nm. This is an unprecedented use of an enzyme nitrate reductase and capping peptide in the synthesis of Nd_2_Se_3_ NPs. Lastly, these investigations will help assist in developing a rational enzymatic approach for the fabrication of NPs with a variety of compositions as well as tunable surface morphologies.

## 2. Materials and Methods

All the chemical reagents were acquired from commercially available sources and were of the highest purity available. The enzyme nitrate reductase was purified as described by Kumar et al. [18].

### 2.1. Green Synthesis of Nd_2_Se_3_ NPs

The reaction mixture (3 mL), containing NdCl_2_ (1 mM), SeCl_4_ (1 mM), NaNO_3_ (5 mM), 100 μg (2.66 × 10^−2^ mM) of synthetic peptide having the amino acid sequence (Glu-Cys)n-Gly, 1.0 mM α-NADPH (a reduced form of nicotinamide adenine di-nucleotide phosphate sodium salt), and 1.66 U of nitrate reductase, was incubated under anaerobic conditions at 25 °C. After 12 h, the reaction mixture was exposed to UV-Vis spectrophotometric measurements. On completion of the reaction, Nd_2_Se_3_ NPs were subjected to treatment with 50% *v*/*v* of ethanol to eliminate unbound proteins and were then used for further characterization. Eventually, NPs were precipitated at 30,000 g for half an hour.

### 2.2. Characterization of Green Synthesized Nd_2_Se_3_ NPs

The confirmation of synthesis was ascertained utilizing UV-Vis spectrophotometer from Shimadzu (UV-1601 PC). Fluorescence was performed by exciting the samples at 330 nm, and the emission spectra were logged from 400 to 700 nm using a spectrofluorometer, FLS920, Edinburgh Instruments, UK at a scan rate of 300 nm/min. X-ray diffraction (XRD) patterns were logged in the 2θ range of 20°–80° with a step size of 0.02° and 5 s per step using a Philips X’PERT PRO armed with X’celerator, a rapid solid-state detector with iron-filtered Cu Kα radiation (λ = 1.5406 Å) as the source. The Nd_2_Se_3_ NP suspension coated onto carbon coated copper grids was subjected to TEM analysis using an FEI Tecnai 30 TEM operated at 300 kV. The FTIR spectroscopy of bioengineered Nd_2_Se_3_ NP was performed in KBr pellets using a Perkin-Elmer Spectrum One instrument. The spectrometer was operated in the diffuse reflectance mode at a resolution of 2 cm^−1^. To obtain good signal to noise ratio, 128 scans of the film were taken in the range of 450–4000 cm^−1^. The dried powder of bioengineered Nd_2_Se_3_ NPs was used for thermogravimetric analysis on a Q5000V 2.4 Build 223 instrument by applying a scan rate of 10 °C min^−1^ [26].

## 3. Results

The schematic representation of biosynthesis and proposed mechanism of Nd_2_Se_3_ NPs is depicted in Figure 1.

The present study examined the modified and advanced processes for the greener production of nanomaterials using enzyme nitrate reductase purified from the extra cellular broth of *Fusarium oxysporum*, as reported by Kumar et al. [18]. After incubation at 25 °C for 12 h under anaerobic circumstances, the reaction mixture (3 mL) in 200 mM phosphate buffer (pH 7.2) containing freshly prepared NdCl_2_, SeCl_4_, NaNO_3_, synthetic peptide (Glu-Cys)n-Gly, NADPH, and nitrate reductase resulted in the creation of Nd_2_Se_3_ NPs (Figure 1). The synthesis was monitored by following the attendance of the absorption band centered at 330 nm, which indicated the formation of Nd_2_Se_3_ NPs [27]. The absorption band at 270–280 nm is visible in Figure 2a, which is attributable to the proteins and α-NADPH used in the reaction. The suspension was very stable, with no sign of aggregation of the NPs even after one month. No absorption band at 330 nm was found without or denatured nitrate reductase or NADPH (data not shown). This observation confirmed the involvement of the enzyme in the oxidation of neodymium (from Nd^2+^ to Nd^3+^) and reduction of selenium (Se^4+^ to Se^2+^) to produce Nd_2_Se_3_ NPs utilizing nitrate as a substrate and NADPH as a cofactor. We used the fungus *Fusarium oxysporum* for the extracellular biosynthesis of Nd_2_Se_3_ NPs; the same had been used in earlier studies to purify the nitrate reductase (18).

Eventually, the optical energy band gap of Nd_2_Se_3_ NPs was estimated via the Tauc Equation (1) shown below
αhν = A (hν − E_g_)^1/n^
(1)
where α, A, h, and ν are absorption co-efficient, arbitrary constant, Plank constant, and light frequency, respectively. Further, on the basis of electronic transition, the n-parameter could be 1/2 (direct allowed), 2 (indirect allowed), or 3 (indirect forbidden) [28].

The absorption co-efficient (α) was estimated from the UV-Vis absorbance parameters via the following Equation (2),
α = 2.303 (A_b_/t)(2)
where A_b_ is the absorbance of nanocrystals and t is the thickness of the cuvette used for measurements. The optical band gap of a nanocrystals was obtained by extrapolation of the linear region of the plot (αhν)^2^ vs. hν to the point (αhν)^2^ = 0 [29]. The calculated band gap for Nd_2_Se_3_ NPs was found to be 3.75 eV, and is shown in Figure 2b.

Fluorescence spectrum of Nd_2_Se_3_ NPs was obtained at 330 nm excitation, as depicted in Figure 2. Whereas emission spectra revealed a band at 421 nm with red shift (Figure 3). The emission band at 421 nm is considerably red shifted compared to its absorption onset, which is credited to the band-gap or near-band-gap emission. The small shift in the peak suggests that the NPs possess a continuous surface with most surface atoms exhibiting the coordination and oxidation states of their bulk counterparts. The full width at half maximum (FWHM), which is the extent of the spread of peak, is 88.9 nm.

The spherical shape and size (18 ± 1 nm) of the Nd_2_Se_3_ NPs were confirmed by TEM (Figure 4a) and the spherical formation of the nanocrystals may have been due to their dynamic nature. To further verify the crystallinity of Nd_2_Se_3_ nanoparticles, the diffraction patterns of X-ray was recorded from drop cast films of biogenic Nd_2_Se_3_ nanoparticles. The as-synthesized Nd_2_Se_3_ nanoparticles were found to reveal the crystalline nature owing to well-defined Bragg’s reflections. The peak position and 2θ values agree with those reported for Nd_2_Se_3_ nanoparticles, almost all peaks in the pattern could be indexed to cubic phase cell parameters, a = b = c = 8.85, α = β = γ = 90° [JCPDF # 190823]. The given investigation has confirmed the first ever synthesis of Nd_2_Se_3_ nanoparticles under ambient conditions. The XRD pattern of the Nd_2_Se_3_ NPs showed intense peaks at (211), (220), (310), (321), (420), (422), (521) (611), (620), and (444) in the 2θ range of 20°–80° (Figure 4b) and agrees with those reported for the Nd_2_Se_3_ nanocrystals. The size of particles under XRD was found to be ~16 nm (Figure 4b). Further, the confirmation of encapsulation of proteins over the surface of particles was realized through FTIR. Therefore, the peak at 1635.56 cm^−1^ corresponds to characteristic of C=O of amide group of the amide I linkage confirmed the presence of protein. Further, the peak at 3320.62 cm^−1^ confirms the N-H stretching vibration. The C-N stretching of aliphatic amines associated with peptide bond was affirmed by extra characteristic peak at 1044.54 cm^−1^ (Figure 4c).

The gravimetric analyses were performed to check the thermal stability of capping agents. TGA was performed on as-synthesized Nd_2_Se_3_ NPs in a nitrogen gas environment at temperatures ranging from 20 °C to 800 °C to calculate the amount of proteins present on the nanoparticles. As pointed out earlier, the as-synthesized nanoparticles are capped with proteins that stabilize them against aggregation. As a result, it degrades in two stages. Weight loss occurs up to 120 °C due to the evaporation of adsorbed water. In the second stage, loss is attributed to the decomposition of proteins bound on the surfaces of nanoparticles. The coating was found to contribute almost up to 18% which can be inferred via weight-loss when the particles were heated up to 600 °C. A further increase in the temperature shows a loss of weight that can be accounted for the decomposition of nanoparticles (Figure 4d).

The DTA study was also performed on the same particles. The endothermic weight loss was found to be around 110 °C which indicated the elimination of adsorbed water molecules on the surface of nanoparticles. Again, endothermic weight loss was found around 300 to 400 °C due to the decomposition of proteins (Figure 4d).

Additionally, DLS and zeta potential were performed to check the hydrodynamic radii along with different populations of particles and stability of nanoemulsion. The hydrodynamic radii of particles under DLS were found to be ~57 nm with PDI value of 0.440 (Figure 5A). Zeta potential was found to be −9.47mV (Figure 5B) which indicated the substantially high stability due to low value of Hamaker constant; TEM revealed only the size of the inorganic core, whereas DLS offered the hydrodynamic radii (inorganic core plus hydration layers). The suspension was found to be stable up to 3 months and there were no significant changes in DLS and Zeta potential.

### Possible Mechanism

There are several methods to produce semiconductor NPs, but biological methods are rarely used. In the present study, a very simple and effective greener synthesis method for semiconductor NPs was developed. Nd_2_Se_3_ semiconductor NPs were synthesized by the instantaneous oxidation of a metal (from Nd^2+^ to Nd^3+^) and the reduction of nonmetal ions (Se^4+^ to Se^2+^) using purified nitrate reductase enzyme in the presence of the synthetic peptide (Glu-Cys)n-Gly (where n = 5–7), as a capping and stabilizing agent containing repetitive glutamate and cysteine amino acids.

Nitrate reductase has a strong reducing property [30] due to its moderate reduction potential (+0.44 V), which participated in the redox reaction where oxidation of Nd^+2^ into Nd^+3^ took place and simultaneously reduction of selenium from Se^+4^ into Se^+2^ occurred with the synthesis of nano sized materials (Figure 1). During the synthesis, the synthetic peptide [(Glu-Cys)n-Gly, where n = 5–7] served as a stabilizing agent, which not only reduced steric hindrance and static-electronic repulsive forces between metal and nonmetal, but also served as a capping agent responsible for attaining the required sizes and shapes. Synthetic peptides comprising repetitive units of glutamate [31] and cysteine [32] amino acids were designed by virtue of their tendency to interact with inorganic NPs. The designed peptides stabilize semiconductor Nd_2_Se_3_ NPs and enable them to interact with various molecules due to the existence of different functional groups. The interaction of peptides and NPs depends upon the chemical properties of NPs, peptides, and reaction parameters and the dynamics of their interaction is responsible for the long-term stability of peptide-capped NPs. The electronic properties of NPs can be perfectly determined by UV-Vis spectroscopy, and their absorption peak and its width is directly correlated to the chemical composition and size of the particles. The energy of far-UV light is sufficient to excite the electrons of Nd_2_Se_3_ NPs. Therefore, the size and concentration of the Nd_2_Se_3_ NPs were determined by UV-Vis spectroscopy and were further confirmed by DLS (hydrodynamic radii) and TEM (inorganic core) [33].

The overall NP and peptide interaction is a multifunctional phenomenal process, and its properties are determined not only by the characteristics of the NPs, but also by the interacting peptides and the reaction parameters. Particularly, the rate of specific association and dissociation of the involved peptide will determine its longevity in the interaction with the NP surface. Moreover, the synthetic conditions determine the morphological nature and miscibility of a metal and a nonmetal in semiconductor NPs. Generally, semiconductor NPs comprising two dissimilar elements (metal and non-metal) has received greater attention than metallic NPs, both scientifically as well as technologically [34,35]. The blending of two different constituting elements can result in morphological changes in the semiconductor NPs, wherein an extra degree of freedom is developed [36]. However, the constituting elements (metal and nonmetal) and the size determine the behavior of the semiconductor NPs. Mostly, composition and size generally provide the tendency to optimize the energy of the plasmon absorption band of the metal and non-metal blend, which delivers a multipurpose tool for biological applications. Lastly, by means of semiconductor formation, the catalytic nature of the resultant NPs can be enhanced to a reasonable extent, which may not be achievable when employing its corresponding monometallic NPs.

The energy of the incident photon at a wavelength of 300 nm is 4.136 eV, which is responsible for the electronic transitions of Nd_2_Se_3_ NPs. The spectrum obtained describes the chemical composition and particle size. It also confirms the synthesis of the NPs, and their difference from bulk counterparts.

The dependency of the optical nature on particle size is mostly an effect of the internal structure of the nanocrystals. However, as the crystals become smaller, the quantity of involved atoms on the surface increases, which greatly influences optical behaviors. If these surface energy states are for nanocrystal band gaps, they can trap charge carriers at the surface and decrease the overlap between the electron and the hole. The number of electrons excited in the conduction band (CB) is a function of the temperature and magnitude of the energy band gap (Eg), defined as the separation between the maximum energy in the valence band and the minimum energy in the CB. As our biogenic material Nd_2_Se_3_ revealed, the optical intensity at 330 nm correlates with 3.75 eV (band gap), it can also be calculated by the following Equations (3) and (4).
E_g_ = 1240/(λ (nm)) (3)
E_g_ = 1240/330 = 3.75 eV (4)

The roles of nano-chalcogenide have been exploited in a variety of newer and emerging technologies. Their applications in broad categories are derived from their unique tunable chemical and physical properties, which give rise to their potential uses in the fields of biomedical, nonlinear optics, luminescence, electronics, catalysis, solar energy conversion, optoelectronics, among others. With decrease in size of nano-chalcogenides, the percent of surface atoms and the value of band gaps increase leading to the surface properties playing an important role in the properties of the materials (14). The transition metal nano-chalcogenides have specific applications. The group II–VI chalcogenide semiconductors have their role in optoelectronic light-emitting diodes and optical devices owing to their wide-bandgap. Additionally, V–VI main-group nano-chalcogenides due to their semiconductor nature have applications in television cameras with photoconducting targets, thermoelectric devices, and electronic and optoelectronic devices and in IR spectroscopy (16). The inner transition metal nano-chalogenides are very rare and their synthesis ought to be exploited in detail because of their non-toxic nature (9) which can be explored for their fluorescence properties toward biomedical applications.

## 4. Conclusions

The present study enabled the development of environmentally friendly biogenic methodology to produce Nd_2_Se_3_ NPs utilizing biologically relevant molecules. Neodymium selenide NPs were synthesized using enzyme nitrate reductase. They were orthorhombic with a size distribution of 18 ± 1 nm (under TEM); the chemical formula was found to be Nd_2_Se_3_ based on the crystal structure of the fabricated NPs. Our ability to fabricate these NPs via a non-toxic, eco-friendly method presents a significant advancement in developing “greener” technique to produce NPs. Lastly, this study offers an alternative approach for the environmentally friendly, cost-efficient, and commercial fabrication of water dispersible Nd_2_Se_3_ NPs. These Nd_2_Se_3_ NPs have favorable bio-medical application potential without any additional requirements for further functionalization.

## Figures and Tables

**Figure 1 biomimetics-07-00150-f001:**
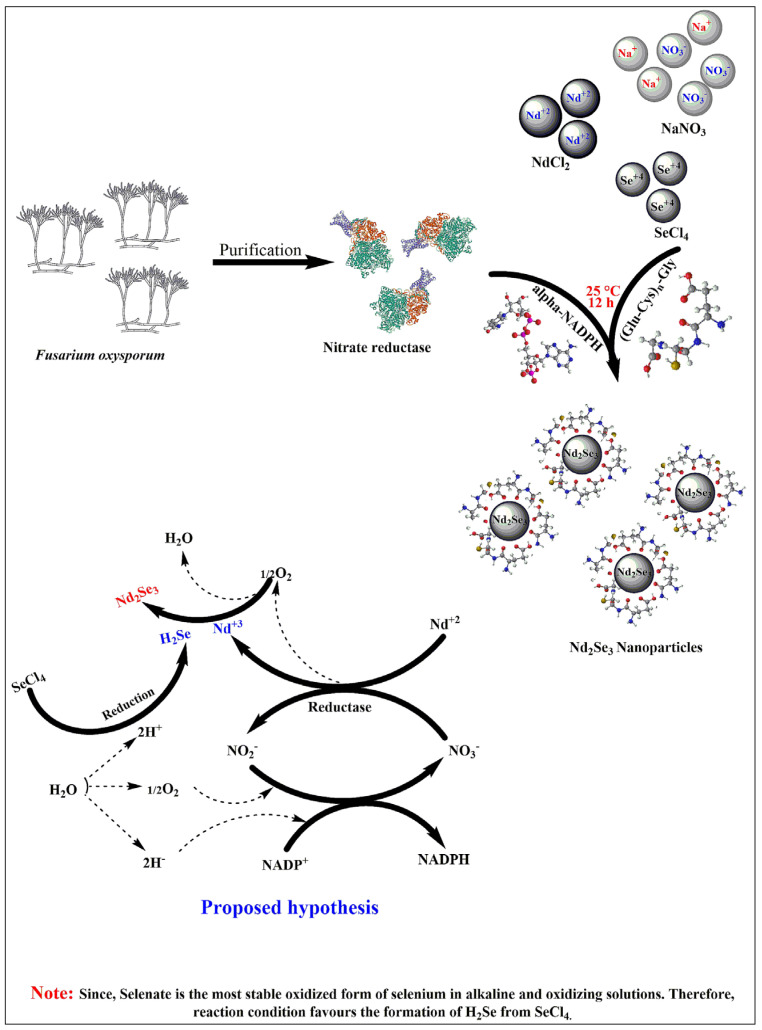
Schematic representation for synthesis of neodymium nanoparticles and proposed mechanism.

**Figure 2 biomimetics-07-00150-f002:**
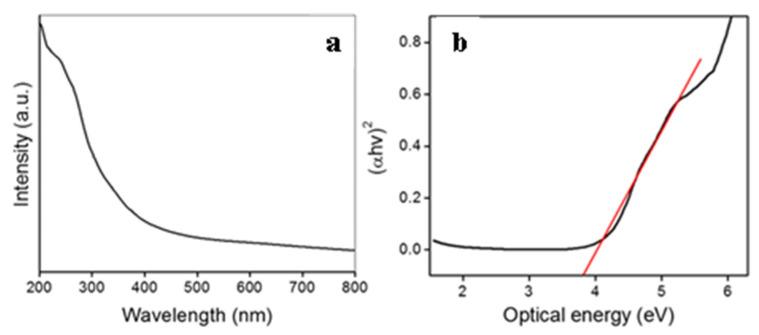
(**a**) UV/Visible absorption spectrum and (**b**) Tauc plot of neodymium selenide nanoparticles.

**Figure 3 biomimetics-07-00150-f003:**
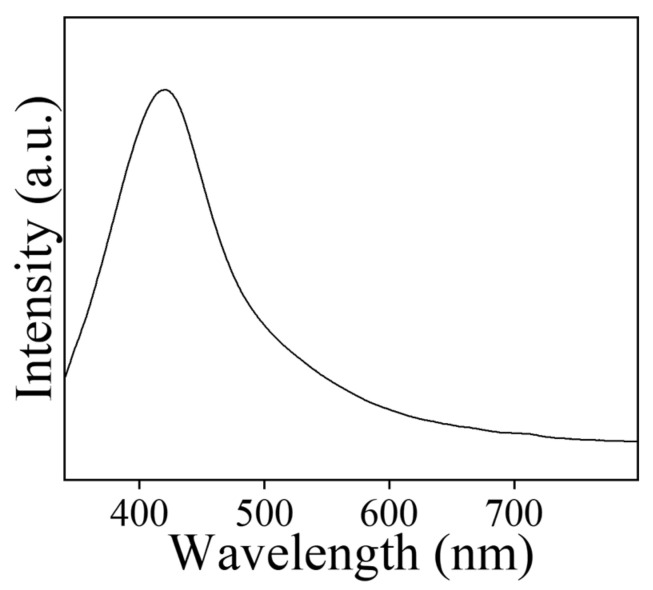
Fluorescence measurement of neodymium selenide nanoparticles.

**Figure 4 biomimetics-07-00150-f004:**
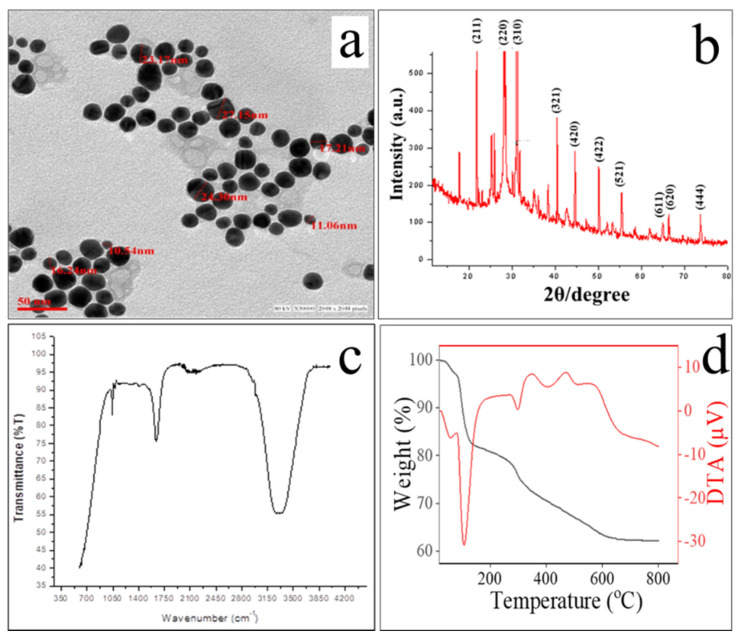
(**a**) TEM, (**b**) XRD, (**c**) FTIR and (**d**) TGA/DTA analysis of biogenic Nd_2_Se_3_ NPs.

**Figure 5 biomimetics-07-00150-f005:**
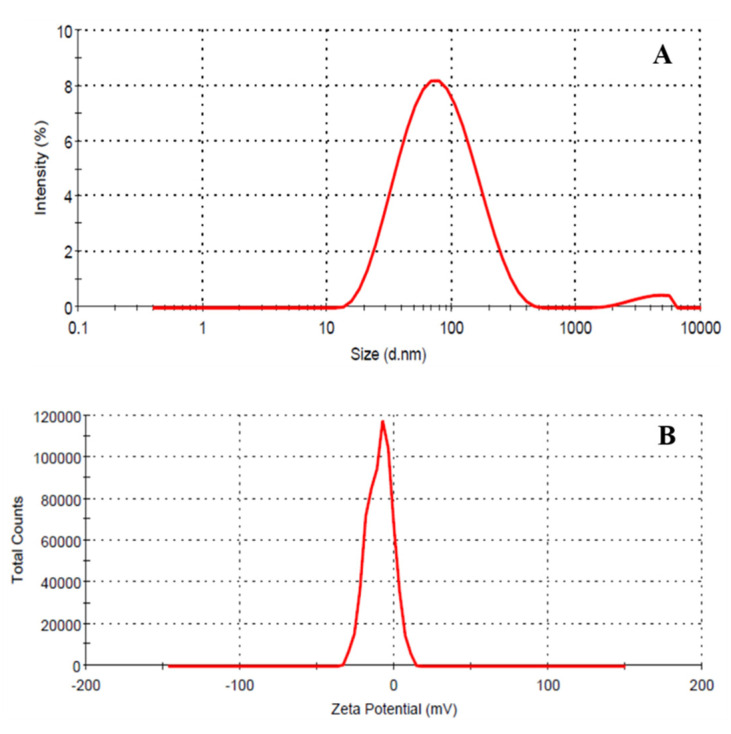
(**A**) DLS and (**B**) zeta potential for Nd_2_Se_3_ NPs.

## Data Availability

Not applicable.

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
