# Peer review of "Neodymium Selenide Nanoparticles: Greener Synthesis and Structural Characterization"

_biomimetics, 2022, doi:10.3390/biomimetics7040150_

Round 1

Reviewer 1 Report

 Ansary et al., have reported on the synthesis and characterisation of Neodymium Selenide nanoparticles. This manuscript is organised and written well. however, there are a few issues that need to be addressed before suggesting for publication.

-          Figure2, panel A and B are not explained in the caption.

-          The scale bar on Figure 4a is not clear

-          For the sake of consistency, could the authors either use lower and capital letters in the figures

-          Could the authors provide the average particle size distribution plot?

-          There is no indication that this is an in vitro synthesis i.e., biocompatibility test etc

-          it is perhaps better to call it green or eco-friendly fabrication/synthesis of Neodymium Selenide nanoparticles as using in vitro could be misleading.

Author Response

Response to the Reviewers

We thank reviewers and editors for their valuable suggestions to improve my manuscript and bring it to the standard of the journal. We have strived to resolve all the queries raised by reviewers, if any further changes are required, please notify us.

Reviewer 1:

Ansary et al., have reported on the synthesis and characterisation of Neodymium Selenide nanoparticles. This manuscript is organised and written well. however, there are a few issues that need to be addressed before suggesting for publication.

Comment 1#:  Figure2, panel A and B are not explained in the caption.

Response 1#: The corrections have been made and are now incorporated in the revised MS.

Comment 2#: The scale bar on Figure 4a is not clear.

Response 2#: Sharpness of the figure has been improved and are now incorporated in the revised MS.

Comment 3#: For the sake of consistency, could the authors either use lower and capital letters in the figures.

Response 3#: The corrections have been made and incorporated in the revised MS.

Comment 4#: Could the authors provide the average particle size distribution plot?

Response 4#: Average particle size distribution is the best way to analyse different populations in a sample. However,  we are unable to provide particle size distribution plot due to some technical reasons and non-availability. Though, TEM micrograph is highly relevant and confirms the presence of different particles along with average size and we request you to kindly consider this as a proof.

Comment 5#: There is no indication that this is an in vitro synthesis i.e., biocompatibility test etc.

Response 5#: Actually, the synthesis was performed by using purified enzyme and peptide. Therefore, the term in vitro was used. But as per your suggestion “greener synthesis” term has been used now. The in vitro (cell biology) or any other biological activities were not performed for the given work.

Comment 6#: it is perhaps better to call it green or eco-friendly fabrication/synthesis of Neodymium Selenide nanoparticles as using in vitro could be misleading.

Response 6#: As per your suggestion “greener synthesis” term has been used now and incorporated in the MS.

Reviewer 2 Report

The paper by Dr. Ansary  et al. is focused on the synthesis of Nd2Se3 semiconductor nanoparticles, with possible applications in many technological fields, including biomedical. The interesting aspect of this research is the use of a reductase obtained by Fusarium oxysporum as a green alternative to usual synthetic procedures; moreover, the NP have been coated with a peptide to prevent aggregation and supply "chemical handles" for further functionalization.

The paper is well written and the overall characterization well conceived, although a few aspects are to be improved before publication. 

- I would suggest to briefly describe the common synthetic procedures for Nd2Se3 NP in the Introduction, or al least add a few references.

- In line 68 I guess "stroke" is a mistake.

- Under paragraph 2.1, I would suggest to indicate the molar concentration corresponding to the amount of synthetic peptide used in the synthesis; moreover, I would suggest to explain the purification procedure in more detail (i.e. are the NP ultracentrifuged after alcohol washings? Are they lyophilized in the end?).

-In Figure 1 the conversion of SeCl4 to H2Se is labeled as "oxidizing conditions", while selenium undergoes reduction.

- In line 147 I would suggest to replace "circumstances" with "conditions".

- In line 153 "solution" must be replaced with "suspension".

- In line 156 the Authors write "..the involvement of the enzyme in the reduction of Nd2+....", while Nd2+ undergoes oxidation to Nd3+. The same applies to line 251.

- In line 221 the Authors state that, on the basis of TGA results, the peptidic coating contributes to the total weight of the NP for up to 38%; actually at 120 °C the approx. 17% weight loss is due to adsorbed water, and, as at 600 °C the total weight loss is approx. 35%, I guess the coating corresponds to 18% w/w. Could the Authors explain?

- Could the Authors explain in more detail how they could infere the NP chemical composition on the basis of the crystal structure, as stated in the Conclusions?

- As the peptide is been added as a stabilizing agent, I would suggest to include in the paper some stability data, in particular referring to NP dimensions.

Finally, I would suggest the Authors to check English wording, spelling and consistency throughout the paper (e.g. lines 174, 199, 280, 310, etc.).

Author Response

Response to the Reviewers

We thank reviewers and editors for their valuable suggestions to improve my manuscript and bring it to the standard of the journal. We have strived to resolve all the queries raised by reviewers, if any further changes are required, please notify us.

Reviewer 2:

The paper by Dr. Ansary  et al. is focused on the synthesis of Nd2Se3 semiconductor nanoparticles, with possible applications in many technological fields, including biomedical. The interesting aspect of this research is the use of a reductase obtained by Fusarium oxysporum as a green alternative to usual synthetic procedures; moreover, the NP have been coated with a peptide to prevent aggregation and supply "chemical handles" for further functionalization.

The paper is well written and the overall characterization well-conceived, although a few aspects are to be improved before publication. 

Comment 1#:  I would suggest to briefly describe the common synthetic procedures for Nd2Se3 NP in the Introduction, or al least add a few references.

Response 1#:  The classical reference for the synthesis of Nd2Se3 NP,  id Ref no. 27, which is already added. 

Comment 2#:  In line 68 I guess "stroke" is a mistake.

Response 2#: Though stroke can be removed but if we keep it in the text, it will reflect more accurate information. Stokes shift is the difference (in energy, wave number or frequency units) between positions of the band maxima of the absorption and emission spectra (fluorescence and Raman being two examples) of the same electronic transition. 

Comment 3#:  Under paragraph 2.1, I would suggest to indicate the molar concentration corresponding to the amount of synthetic peptide used in the synthesis; moreover, I would suggest to explain the purification procedure in more detail (i.e. are the NP ultracentrifuged after alcohol washings? Are they lyophilized in the end?).

Response 3#: The concentration in terms of “mM” has been incorporated in the revised MS.

Comment 4#:  In Figure 1 the conversion of SeCl4 to H2Se is labeled as "oxidizing conditions", while selenium undergoes reduction.

Response 4#:  The corrections have been made and incorporated in the revised figure 1 of the revised MS.

Comment 5#:  In line 147 I would suggest to replace "circumstances" with "conditions".

Response 5#: The corrections have been made and incorporated in the revised MS.

Comment 6#:  In line 153 "solution" must be replaced with "suspension".

Response 6#: The corrections have been made and incorporated in the revised MS.

Comment 7#:  In line 156 the Authors write “the involvement of the enzyme in the reduction of Nd2+....", while Nd2+ undergoes oxidation to Nd3+. The same applies to line 251.

Response 7#: The corrections have been made and incorporated in the revised MS.

Comment 8#:  In line 221 the Authors state that, on the basis of TGA results, the peptidic coating contributes to the total weight of the NP for up to 38%; actually at 120 °C the approx. 17% weight loss is due to adsorbed water, and, as at 600 °C the total weight loss is approx. 35%, I guess the coating corresponds to 18% w/w. Could the Authors explain?

Response 8#: The typographical error has been corrected and incorporated in the revised MS.

Comment 9#:  Could the Authors explain in more detail how they could infere the NP chemical composition on the basis of the crystal structure, as stated in the Conclusions?

Response 9#:  X-ray diffraction provides phase composition identification and can distinguish the major, minor, and trace compounds present in a sample. XRD analysis includes the mineral name of the substance, chemical formula, crystalline system, and reference pattern number from the ICDD International database. Standardless quantitative information can also be obtained from XRD using Rietveld Analysis.

Comment 10#:  As the peptide is been added as a stabilizing agent, I would suggest to include in the paper some stability data, in particular referring to NP dimensions.

Response 10#:  We have performed DLS and zeta potential after every month for 3 months and there was no substantial change in the values. Therefore, no data was added.

Comment 11#:  Finally, I would suggest the Authors to check English wording, spelling and consistency throughout the paper (e.g. lines 174, 199, 280, 310, etc.).

Response 11#:    We have strived to improve the English.